# A Temperature Prediction Model for Flexible Electronic Devices Based on GA-BP Neural Network and Experimental Verification

**DOI:** 10.3390/mi15040430

**Published:** 2024-03-23

**Authors:** Jin Nan, Jiayun Chen, Min Li, Yuhang Li, Yinji Ma, Xuanqing Fan

**Affiliations:** 1Institute of Solid Mechanics, Beihang University (BUAA), Beijing 100191, China; nanjin@buaa.edu.cn (J.N.); liyuhang@buaa.edu.cn (Y.L.); 2International Innovation Institute, Beihang University (BUAA), Hangzhou 310023, China; chenjiayun@buaa.edu.cn; 3Tianmushan Laboratory, Xixi Octagon City, Yuhang District, Hangzhou 310023, China; 4Laboratory of Flexible Electronics Technology, Tsinghua University, Beijing 100084, China; mayinji@tsinghua.edu.cn; 5Applied Mechanics Laboratory Ministry of Education People’s Republic of China (AML), Department of Engineering Mechanics, Tsinghua University, Beijing 100084, China

**Keywords:** flexible electronic devices, GA-BP neural network, thermal safety, temperature prediction model

## Abstract

The problem that the thermal safety of flexible electronic devices is difficult to evaluate in real time is addressed in this study by establishing a BP neural network (GA-BPNN) temperature prediction model based on genetic algorithm optimisation. The model uses a BP neural network to fit the functional relationship between the input condition and the steady-state temperature of the equipment and uses a genetic algorithm to optimise the parameter initialisation problem of the BP neural network. To overcome the challenge of the high cost of obtaining experimental data, finite element analysis software is used to simulate the temperature results of the equipment under different working conditions. The prediction variance of the GA-BPNN model does not exceed 0.57 °C and has good robustness, as the model is trained according to the simulation data. The study conducted thermal validation experiments on the temperature prediction model for this flexible electronic device. The device reached steady state after 1200 s of operation at rated power. The error between the predicted and experimental results was less than 0.9 °C, verifying the validity of the model’s predictions. Compared with traditional thermal simulation and experimental methods, this model can quickly predict the temperature with a certain accuracy and has outstanding advantages in computational efficiency and integrated application of hardware and software.

## 1. Introduction

Flexible electronic devices have the ability to overcome the limitations of traditional rigid devices that cannot be bent or stretched. By integrating polymer matrix, stretchable wires, and functional components, these devices achieve miniaturisation, lightweighting, and flexibilisation [1,2,3,4,5], making them suitable for a wide range of applications. These include structural health monitoring of spacecraft [6], as well as monitoring blood glucose [7], blood oxygen [8], and electroencephalographic [9] information of flight crews. Due to the complexity and variability of the operating environment of flexible electronic devices, the devices may suffer from short-circuit or power overload problems, and the resulting abnormal heating of the devices poses a challenge to the safety of the mission [10,11,12]. In order to ensure the thermal safety of flexible electronic devices in practical applications, researchers are using a variety of temperature prediction methods.

Currently, temperature prediction methods for flexible electronic devices are based mainly on theoretical calculations and simulation experiments. The theoretical calculation method is based on Fourier heat transfer theory, which is combined with boundary conditions and initial conditions under different environments to solve the heat conduction equation and obtain the temperature field equation of the device under the current working conditions [13,14,15]. The theoretical calculation can obtain the exact temperature distribution of the equipment according to the physical equation, but the solution of the differential equation greatly increases the difficulty of temperature prediction and can only be applied to the simple and ideal application conditions of flexible electronic equipment. Simulation experiments use finite element analysis or environmental testing to calculate the physical state of the equipment, which can quickly analyse the performance of the equipment while ensuring accuracy. However, different operating conditions require separate analysis, resulting in high computational and experimental costs [16,17,18,19,20,21]. Although these methods can effectively evaluate the thermal safety of flexible devices, they also have some problems, such as large computational resources, poor generalisation of prediction ability, and the inability to perform real-time prediction analysis, which lead to the challenge of real-time temperature prediction of flexible electronic devices under different operating conditions.

With the rapid development of artificial intelligence technology, machine learning, as the core of artificial intelligence, has made continuous breakthroughs in the field of algorithms and applications [22,23,24,25,26,27,28,29]. At present, machine learning algorithms mainly include decision tree algorithm, support vector machine algorithm, artificial neural network algorithm, deep learning algorithm, etc. [30,31,32,33,34,35]. Researchers have applied the machine learning method to the field of flexible electronics by combining the characteristics of the algorithm and have achieved good results. The applications of machine learning in the field of flexible electronics mainly include data analysis and understanding of flexible sensors, multimodal information processing and decoupling, and intelligent environmental perception [36]. For example, Fang et al. [37] developed a textile triboelectric sensor system that can realise dynamic cardiovascular condition assessment and high-fidelity pulse monitoring. The system takes the signal collected by the flexible sensor as input and uses the machine learning algorithm to extract the pulse features from the data. The output blood pressure data are obtained by training the neural network. Kim et al. [38] developed a human tactile recognition system based on a flexible array piezoelectric sensor. The system uses a piezoelectric sensor system to obtain multimodal tactile information about materials and establishes the relationship between sensor signals and material types by training a hybrid neural network.

Although machine learning methods have been successfully applied to flexible electronics, there are still some challenges and shortcomings. At present, researchers focus more on using machine learning methods to process the signals of flexible electronic devices, and there is little research on using machine learning methods to improve the thermal safety level of flexible electronic devices. Currently, the thermal safety of flexible electronic devices can be enhanced through active structural design and material design [39]. To ensure thermal safety, flexible electronic devices need to monitor the temperature of certain key points to ensure that the actual temperature is within the design temperature range. The design temperature of the device is related to the actual working condition, so the key is how to predict the design temperature of the device with the least amount of computational resources while ensuring fast and high accuracy. To address this issue, we propose a temperature prediction model for flexible electronic devices based on a GA-BP neural network. The model utilises a BP neural network to establish a mapping relationship between input conditions and predicted temperature. Additionally, we incorporate a genetic algorithm to optimise the neural network parameters, thereby enhancing the accuracy of the forecast model. Compared to the experimental and simulation results, this model predicts at a speed 1000 times faster than the traditional method, provided that the prediction variance does not exceed 0.57 °C. This study, based on a machine learning algorithm, aims to use the limited computing resources of flexible devices to evaluate the thermal safety of devices quickly and in real time and to promote the development of flexible electronic devices in the field of human conformal monitoring.

## 2. Temperature Prediction Model and Experiment

### 2.1. Research Object

At present, the thermal safety problem of flexible electronic technology is becoming more and more prominent. In order to improve the safety of flexible electronic devices, it is an efficient way to predict the thermal physical state of the devices. The modular performance test unit (MPTU) is designed to investigate and predict the thermal safety of various types of flexible electronic devices under different operating conditions. Figure 1 shows the overall structure of the MPTU. Among them, Heater 1 and Heater 2 are two flexible heaters, and Sensor 1 and Sensor 2 are two flexible temperature sensors. These components are firmly attached to the central flexible substrate material, such as PDMS or Ecoflex, with the degree of adhesion regulated by the surrounding bolts. To ensure the thermal safety of flexible electronic devices in practical applications, researchers use a variety of methods to predict the temperature of the devices. Heater, sensor, and substrate material are the three components of flexible electronic devices, and their performance under different working conditions will directly affect the temperature distribution of the whole device. Therefore, during testing, MPTU can assess the thermal safety level of the corresponding type of flexible electronic device by using different heaters, sensors, and substrate materials.

### 2.2. Temperature Prediction Model

#### 2.2.1. Genetic Algorithm

The genetic algorithm (GA) is a heuristic global search method that mimics the process of biological genetic evolution. It encodes the optimisation parameters to shape the individuals of the population with the aim of finding the maximum fitness function. In this process, genetic selection, crossover, and variation are incorporated to screen individuals. By retaining and passing on individuals with good fitness while removing those with poor fitness, the fitness of each generation’s population is constantly enhanced.

The basic steps of the genetic algorithm can be divided into:1.Selecting operationThis operation involves the identification of the dominant individual from the parent population to the offspring population, with the aim of retaining exceptional individuals. A range of selection methods exist, including the roulette and tournament methods. In these methods, the likelihood of selecting the dominant individual is linked to its fitness value, with higher fitness resulting in a higher probability of selection.2.Cross operationThe cross operation involves selecting two individuals from the paternal population and exchanging two chromosomes to create a superior individual. The process of cross operation entails the arbitrary pairing of individuals in a given population, with one or more chromosomal positions being randomly selected for each pair.3.Mutation operationMutation involves selecting an individual from its parent and selecting a specific point within the chromosome to be altered, ultimately creating a more optimally adapted individual.

#### 2.2.2. GA-BP Neural Network Algorithm

The BP neural network is a type of multi-layer feed-forward neural network characterised by forward transmission of the signal and backward propagation of the error. In the forward transmission process, the signal is processed layer by layer, starting from the input layer, moving through the hidden layer, and eventually reaching the output layer. If the output layer does not achieve the expected output, the error is propagated backwards. The BP neural network fine-tunes its network weight and threshold according to the prediction error to ensure that its prediction output continuously approximates the expected output. Figure 2 shows the topology of the BP neural network.

This network is capable of fitting the non-linear functional relationship, where its input and output values represent the independent and dependent variables of the function, respectively. The capacity for associative memory and prediction is achieved through the establishment of a functional mapping relationship between input and output values, which must undergo training. The training process of the BP neural network consists of the following stages:1.The output calculation of hidden layer and output layerAccording to input variable *X*, connection weight *ω_ij_* between input layer and hidden layer, and hidden layer threshold *a*, the output *H*_1,*j*_ of the first hidden layer is calculated.
(1)$$H_{1,j}=f\left(\sum_{i=1}^{n}\omega_{ij}x_{i}-a_{j}\right)\;j=1,2,\ldots ,p_{1}$$
where *p*_1_ represents the number of nodes in the first hidden layer and *f* is the activation function for that layer. *p_t_* denotes the number of nodes in the hidden layer of layer *t*, while *ω_ij_* refers to the weight of the connection between layer *t* and layer *t* − 1. Consequently, the output of the hidden layer of layer *t* can be expressed as:(2)$$H_{t,j}=f\left(\sum_{i=1}^{n}\omega_{ij}H_{t-1,j}-a_{j}\right)\;j=1,2,\ldots ,p_{t}$$Using the output of the last hidden layer *H_j_*, connect the weight *ω_jk_* and threshold *b* to calculate the BP neural network and predict the output *O*:(3)$$O_{k}=\sum_{j=1}^{q}H_{j}\omega_{jk}-b_{k}\;k=1,2,\ldots ,m$$2.Update parameters according to errorsCalculate the network prediction error *e* based on the network prediction output *O* and the expected output *Y*:(4)$$e_{k}=Y_{k}-O_{k}\;k=1,2,\ldots ,m$$After computing the prediction error, it is propagated backwards and subsequently used to update the connection weights *ω* from the output to the input layer:(5)$$\begin{array}{l} \omega_{jk}=\omega_{jk}+\eta H_{j}e_{k}\;j=1,2,\ldots ,p_{t}\\ \omega_{ij}=\omega_{ij}+\eta H_{j}\left(1-H_{j}\right)x\left(i\right)\sum_{k=1}^{m}\omega_{jk}e_{k}\;i=1,2,\ldots ,n;j=1,2,\ldots ,p_{1} \end{array}$$3.The training data are constantly supplied to the network, and the algorithm continues to iterate until the termination criteria are fulfilled, thus accomplishing the network’s training process.

The BP neural network has excellent non-linear mapping capability, but the merit of the initial connection weights and thresholds of the network nodes will affect the training speed and prediction accuracy of the model. Thus, the genetic algorithm’s superior global search capability allows for the optimisation of the initial weights and thresholds of the BP neural network to enhance its predictive output accuracy. The GA-BPNN, composed of the optimised BP neural network with the genetic algorithm, exhibits better convergence capability than the original BP neural network.

The neural network training process consists of two main parts: BP neural network parameter optimisation using the GA algorithm and BP neural network training. Figure 3 illustrates the training process. In the actual process, the population is initialised according to the network structure, and the individual is encoded. Secondly, the fitness function of the individual is constructed. The population was created through selective breeding, crossover, and mutation operations, and iterative evolution was used to obtain a population that meets the termination requirements. The BP network is then initialized based on the optimal individual output before being trained.

### 2.3. Design of Prediction Methods

#### 2.3.1. Thermal Simulation and Experiment

In order to ensure the reliability of thermal simulation experiments and model predictions, the MPTU was selected as the research object in this paper to test the thermal performance of flexible substrates at room temperature. The MPTU test facility used PDMS and Ecoflex (*k*_PDMS_ = 0.4 W/(m∙K), *k*_Ecoflex_ = 0.2 W/(m∙K)) as the flexible substrate materials. The MPTU was loaded with a 28 V regulated power supply, and the power of the heater was measured as *p*_heat1_ = 0.364 W and *p*_heat2_ = 0.353 W. The sensing temperatures were determined by collecting the resistance data of the MPTU flex sensors. The experimental results indicate that the device reaches steady state after 1200 s at rated power, and the temperature at the measured point was *T*_Sencor1_ = 42 °C and *T*_Sencor2_ = 41.5 °C. The experimental equipment is shown in Figure 4.

During the thermal analysis experiment, the finite element software Abaqus 2023 was used to simulate the experimental conditions, and the established model was reasonably simplified according to the characteristics of the equipment. The components were connected by contact, a total of 86,212 meshes were divided by sweep, and the steady-state temperature field of the model was calculated by the heat transfer step. The steady-state temperature field of the model was calculated by the heat transfer step, as illustrated in Figure 5. Total time of the thermal simulation calculation is 175 s. The temperature sensor of the device was placed at the tip of the flexible substrate material. Accordingly, the temperature of PDMS and Ecoflex can be read from Figure 5b: *T*_Sencor1_ = 42.2 °C, *T*_Sencor2_ = 41.1 °C.

In comparison with the experimental results, it can be seen that the temperature prediction error of the thermal simulation is less than 0.9% and the time requirement is 14.6% of the experiment time. Importantly, this technique significantly reduces the overall experiment time while upholding a high level of precision. However, it is worth noting that the accuracy of the thermal simulation calculation is closely tied to the computational resources available. In the case of flexible electronic devices, thermal simulation cannot be used because of the limited computing resources of integrated devices. Therefore, it is of great importance to develop a temperature prediction model for MPTUs. This model should be highly accurate, require few resources, and be fast to compute.

#### 2.3.2. Prediction Method Based on GA-BP Neural Network

The GA-BPNN temperature prediction model is built in three steps. Firstly, the training data are created through a batch processing procedure. Afterwards, the data are screened in accordance with the thermal simulation results of the batch calculations to form the final dataset. Secondly, the neural network model is trained using the pre-processed data. Lastly, the model’s performance is evaluated. The specific construction process is illustrated in Figure 6.

Step1: Generate training data

The neural network creates a non-linear mapping between input and output based on extensive training data. The quality of this data impacts the neural network’s prediction performance. Appropriate training data enhances the model’s ability to learn more features using as little data as possible. Since the thermal simulation experiment of the MPTU equipment greatly improves the speed of the experiment compared with the thermal experiment while guaranteeing high accuracy, the training data for the prediction model can be obtained through the batch simulation of the MPTU. As the MPTU can be assembled in a modular fashion using a variety of flexible heaters and flexible substrate materials, experimental thermal simulation data are required at different heating powers and substrate materials. Simulation experiments have been chosen for this purpose. Heater 1 and Heater 2 have heating powers of *P*_Heater1_ and *P*_Heater2_, respectively. The substrate material’s thermal conductivity is given by *k*_1_ and *k*_2_, while the heat transfer coefficients between the substrate material and the flexible heaters are *h*_heater1_ and *h*_heater2_, and those between the substrate material and the sensors are *h*_sensor1_ and *h*_sensor2_. The convective thermal coefficient is denoted as *h*. A total of nine parameters are utilized as variables to compute the MPTU simulation results within the parameter interval. Additionally, there are convective heat transfer coefficients h in the model and environment.

A rational design of experiments (DoE) is necessary for the generation of a significant amount of MPTU simulation data. For efficient exploration of the experimental space, the simulation experiment design must fulfil the requirements of non-collapsing and space-filling. Multiple experimental design methods currently exist, including classical experimental design, uniform design, and Latin hypercube sampling (LHS). LHS is a technique for quasi-random sampling from multivariate parameter distributions that uses stratified sampling to achieve non-overlapping samples and full coverage of the value range through homogeneous stratification and random selection, meeting the requirements of simulation experiment design. In this paper, the LHS methodology is utilised for parameter sampling to construct an optimal sampling space. Additionally, 10,000 sets of initial training data are generated using the software MATLAB 2023b with 9 input parameters. The distributions of some parameters are presented in Figure 7a.

The generation and subsequent analysis of a substantial quantity of simulation cases is a notably time-intensive process. As a result, it is imperative to utilise the Abaqus batch programme for managing the simulation and post-processing tasks. In this paper, an initial training dataset was assembled using the finite element simulation batch technique. The approach initially employs a Python program to read input parameter data produced by MATLAB, which generates corresponding .inp and .bat files. Abaqus software is used to carry out batch simulation calculations through a Windows batch processing script. Conclusively, steady-state temperatures of the two temperature measurement positions in all calculation models are batch exported by reading the .odb file formed by the calculations. This process generates a temperature dataset containing 10,000 sets of simulation calculations.

Due to the non-linear correlation between the input parameters of the model and the temperature, the uniformity of the input parameters in the sampling space does not illustrate an appropriate distribution of the temperature associated with the calculations in this space. The experimental findings of this paper indicate that when the training data obtained from batch processing is applied to model training, the standard deviation of the neural network’s predicted results’ error exceeds 5 °C. This reduces the prediction accuracy below the desired level. The analysis of simulated temperature results reveals that, among 10,000 groups, the lowest temperature recorded is 25 °C, while the highest is greater than 1800 °C. Furthermore, 87% of the available data fall below 100 °C. Notably, the neural network training is impaired by the inhomogeneity of the output parameters, especially in cases of individual and unreasonable high temperature data. As the MPTU experiment’s highest temperature does not surpass 100 °C and the temperature distribution under 100 °C is uniform, data with temperatures below 100 °C were chosen for model training. This collection consists of a total of 8681 training datasets, each containing 9 inputs and 2 outputs (*T*_sensor1_ and *T*_sensor2_, as shown in Figure 5b). The distribution of certain input parameters in the training dataset is illustrated in Figure 7b. Analysis of the input parameter distribution in Figure 7b reveals that certain regions of the modified input parameters are absent, although some of them are still able to maintain the distribution characteristics of LHS sampling.

Step2: Training model

Variations in the physical significance of distinct parameters in the training data of the neural network result in discrepancies in both the value intervals and magnitudes of the data. Consequently, this data comparability issue affects the efficacy of the model’s training. To mitigate this effect, the data can be normalised. The commonly employed normalisation strategies include min–max normalisation, Z-score normalisation, and non-linear normalisation. The min–max approach involves linearly transforming the original data to a set interval to attain isometric scaling of the data. This technique efficiently eradicates adverse effects caused by singular sample data while conserving the distributional characteristics of the sampled data. The paper adopts the min–max normalisation method to normalise all the parameters in the training dataset to the interval −1~1. From the normalised dataset, 90% of the data are selected for network training, and 10% of the data are used to test the network.

The GA-BP neural network temperature prediction model presented in this paper is developed and designed using MATLAB software. The GA-BP neural network structure is set as 9-16-16-2 according to the neural network modelling experience; the initial learning rate is 1 × 10^−3^, the preset mean square error (MSE) training target is 5 × 10^−5^, and the study on the GA population shows that the MSE and prediction error of the GA-BP model are small and gradually converge when the population reaches 200. To ensure computational efficiency, we recommend a population size of 200. After 20 iterations of evolution, the population’s optimal fitness was 1.06. Additionally, the resulting optimized BP neural network meets the convergence requirement, when MSE = 2.81 × 10^−5^ after 244 iterations of training.

Step3: Evaluation of model performance

Once a model has been built, it is important to assess whether its performance meets the prediction requirements. The evaluation process involves two main methods: Measuring model test error and experimental result error. To determine the training level and regression accuracy of the model, MSE, correlation coefficient *R*^2^, and output parameter error distribution are used. MSE is utilised to quantify the variance between the predicted and expected values. The *R*^2^ displays the level of linear correlation between the predicted and expected values, whereby the closer *R*^2^ is to 1, the model’s performance improves. Examining the error distribution of each output parameter helps in assessing the neural network model’s overall prediction efficacy and compares the adaptability of each output parameter to the model.

To assess the efficacy of the GA-BP neural network prediction model, we input the test dataset and experimental data into the model and undertook a comprehensive evaluation. The MSE of this model is 2.81 × 10^−5^, and the deviation from the expectation is within the permissible range. As illustrated in Figure 8, the model underwent regression analysis, resulting in an *R*^2^ greater than 0.999 for the training, verification, and test sets. This analysis satisfies the criteria for practical engineering applications.

Additionally, a statistical analysis was conducted on the error distribution of two output parameters, *T*_sensor1_ and *T*_sensor2_, as demonstrated in Figure 9. The mean error values for *T*_sensor1_ and *T*_sensor2_ are −0.0095 and −0.0055, respectively. The error variance of *T*_sensor1_ is 0.3861, and that of *T*_sensor2_ is 0.5752. The error variances for the two output parameters are similar, indicating that the prediction model has an equally effective prediction for different output parameters. Thus, the network structure is reasonably set. The minor level of error variation is satisfactory for meeting the temperature prediction accuracy criteria of MPTU. Figure 9a demonstrates this by using the output parameters of *T*_sensor1_ as a sample. Following a comparison of the error distribution of its training and test data, the prediction error distribution of both datasets was found to be very similar. This indicates that the model has an equally satisfactory fitting effect on the training and test sets, indicating that there is no problem of overfitting or underfitting.

## 3. Results and Discussion

### 3.1. Comparison of Results

In order to further verify the validity of the neural network prediction model, the experimental conditions are input into the model for calculation. The calculation results are shown in Figure 10. It only takes 0.0124 s for the model to calculate the output *T*_sensor1_ = 42.9 °C and *T*_sensor2_ = 41.8 °C. The error is less than 3% compared with the experimental results, and the output results meet the requirements of MPTU for temperature prediction.

### 3.2. Model Setup Analysis

In order to further explore the predictive ability of the model, this study investigates the following aspects of the proposed neural network model: normalisation method, network structure.

#### 3.2.1. Normalization Method

The neural network predictive model was constructed with identical methods and parameters to those in Section 2.3.2. The predictive performance of the model was evaluated for three normalisation methods: Min–max normalisation to [0, 1], min–max normalisation to [−1, 1], and Z-score normalisation. As illustrated in Figure 11, in comparison to the distribution of model prediction errors for three distinct normalization methods, the most effective technique is the Min–max normalization of training data to [−1, 1]. This approach has the smallest mean error and variance, resulting in superior model prediction performance.

#### 3.2.2. Network Structure

Based on the analysis of this model’s predictive efficacy through normalization methods, this paper has opted for the min–max normalization technique, ranging from [−1, 1], and created a prediction model utilizing similar parameters from Section 2.3.2. The study also explores the correlation between the number of neurons in the model’s hidden layer and its predictive accuracy. As shown in Figure 12, the analysis of nine hidden layer structures reveals that the smallest mean and variance of the model error are present in the hidden layer structure of 16–32. This comparison is based on the distribution of model prediction errors.

#### 3.2.3. Analysis of Results

In summary, the model’s performance is significantly influenced by the normalization method and network structure employed in the prediction model, and further optimization can be made based on the analysis results. The following are the impacts of the normalization method on the model performance: (1) The experimental findings indicate that the min–max normalization approach is better suited for the model than the Z-score method. (2) The min–max normalization reduces the data to various intervals, which has a definite impact on the model’s performance. The model delivers optimal performance when the data ranges between −1 and 1. The model’s performance is influenced by the network’s architecture in the following ways: The experiment reveals no evident correlation between the count of hidden layer neurons and model efficiency. The performance of the model does not inevitably escalate with increasing neurons. Additionally, if the total number of neurons in the hidden layer remains constant, the number of neurons in different layers has a noteworthy impact on model efficacy.

The neural network prediction model utilised in this investigation boasts substantial benefits over conventional experimental and thermal simulation techniques. While maintaining optimal precision, the efficiency of the neural network prediction model is vastly enhanced, with processing speeds 96,000 times faster than experimentation and 14,000 times quicker than finite element simulation methods. After being trained, the neural network model does not require extensive computing resources to run offline and can perform calculations directly.

## 4. Conclusions

This study aims to address the difficulty of evaluating the thermal safety level of different flexible electronic devices, and prepares an experimental device, MPTU, which can modularise the thermal safety level of different flexible electronic devices. A BP neural network temperature prediction model based on genetic algorithm optimisation (GA-BPNN) is proposed for this device. The model can effectively capture the non-linear correlation between the input and output parameters while optimizing the BP neural network’s performance. The model’s rationality was confirmed by analysing the output parameters, and its reliability was tested through experiments and finite element simulation. The findings demonstrate that this study’s model has comprehensively learned the mapping of the relationship between temperature and working conditions. Furthermore, it can accurately predict MPTU experimental outcomes under varying operating conditions, exhibiting both precision and robustness. Compared to conventional experimental and simulation methods, the GA-BPNN temperature prediction model swiftly calculates temperature in various complex working conditions, making it ideal for device integration with minimal computational resource requirements. Furthermore, optimizing the model normalization method and network structure can enhance the model’s prediction accuracy further. This paper considers the influence of material properties and contact relationships of flexible substrates on the safety of devices, which has implications for the selection of flexible packaging/fill materials for the thermal safety design of flexible electronic devices. The neural network model not only improves the efficiency of flexible electronic safety evaluation, but also opens up ideas for the application of other flexible electronic devices. The future application of the deep learning model in the field of flexible electronics will not only be limited to researching the current forward problem but also have a greater application prospect for solving the inverse problem.

## Figures and Tables

**Figure 1 micromachines-15-00430-f001:**
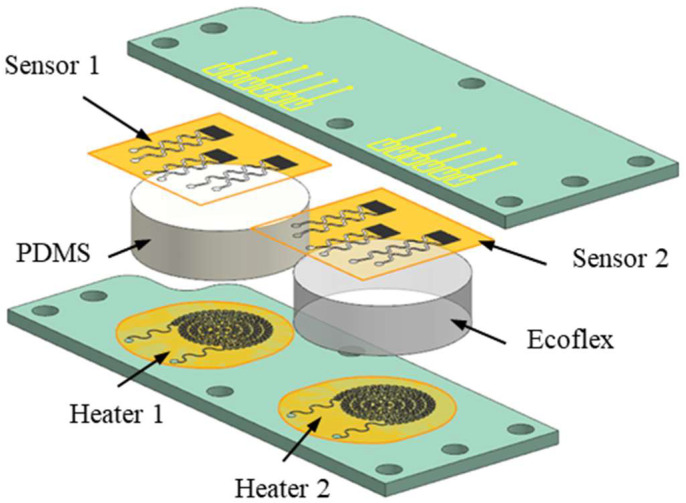
Overall structure diagram of MPTU.

**Figure 2 micromachines-15-00430-f002:**
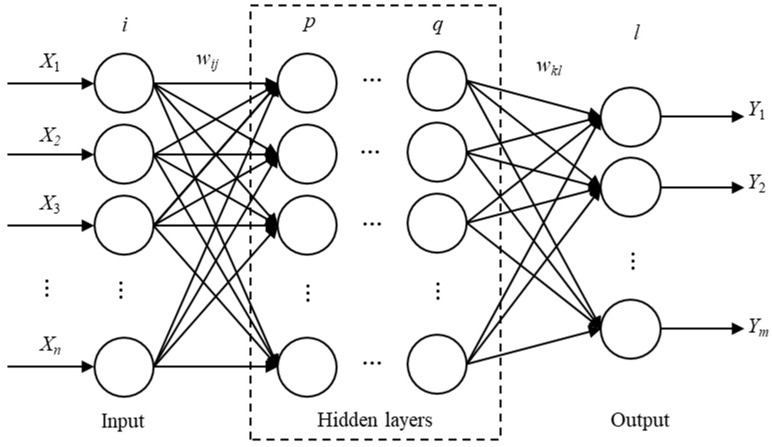
BP neural network topology structure.

**Figure 3 micromachines-15-00430-f003:**
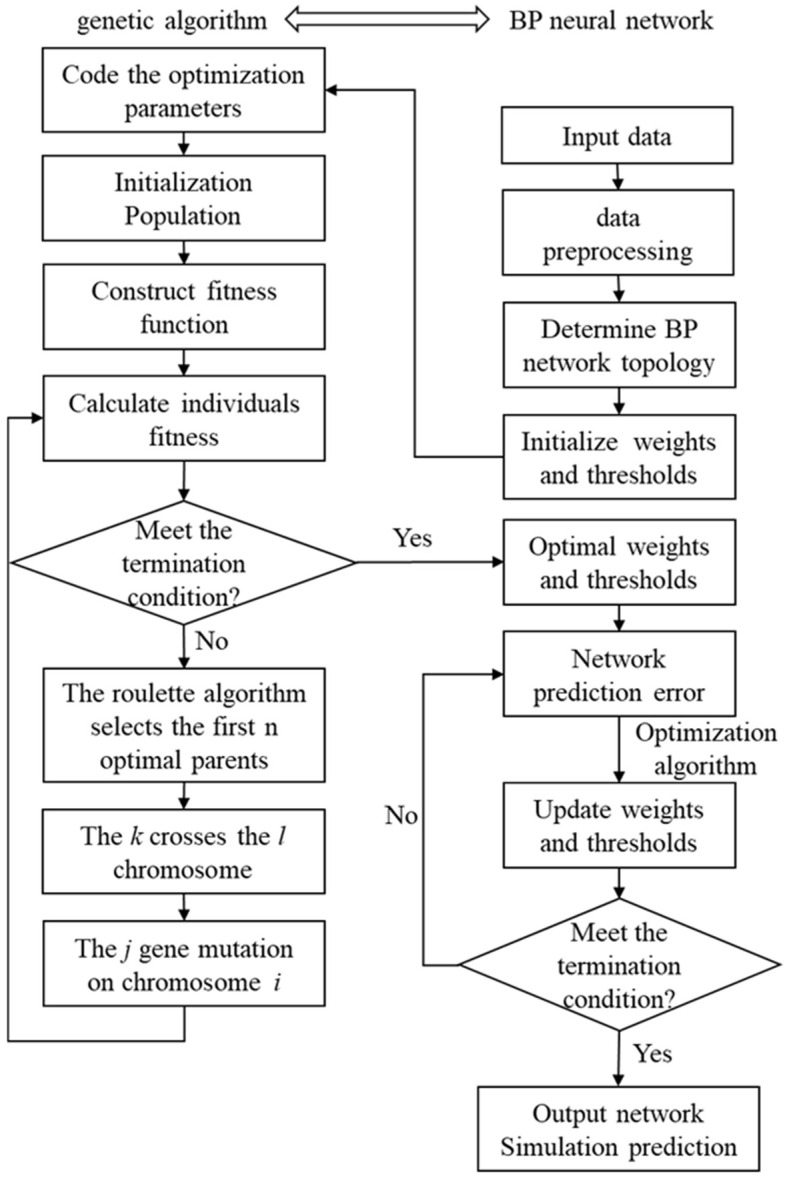
Flow chart of GA-BPNN.

**Figure 4 micromachines-15-00430-f004:**
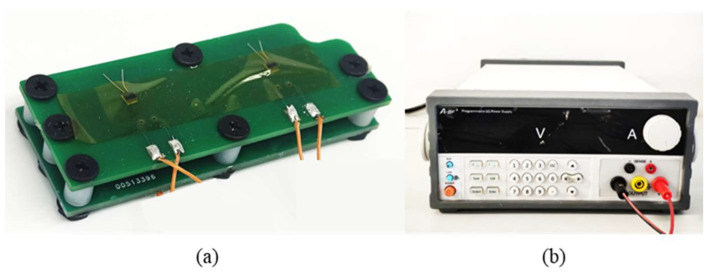
Thermal experimental research object and test equipment. (**a**) MPTU; (**b**) experimental voltage stabilised power supply.

**Figure 5 micromachines-15-00430-f005:**
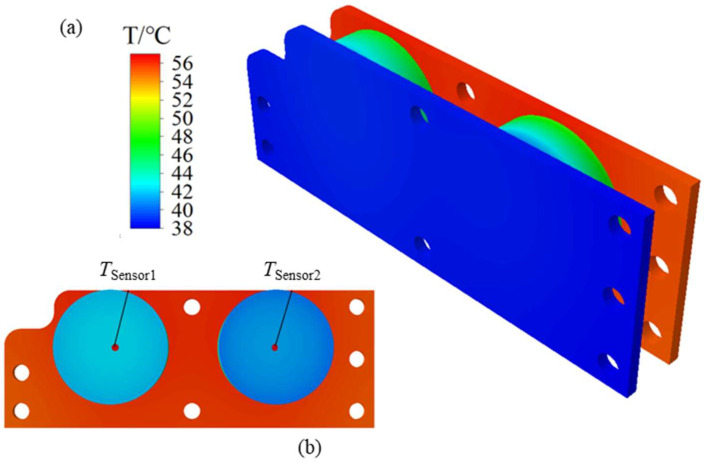
Abaqus FEA result of MPTU for experimental conditions. (**a**) MPTU temperature cloud image; (**b**) temperature distribution near the temperature measuring point (*T*_Sensor1_ and *T*_Sensor2_).

**Figure 6 micromachines-15-00430-f006:**
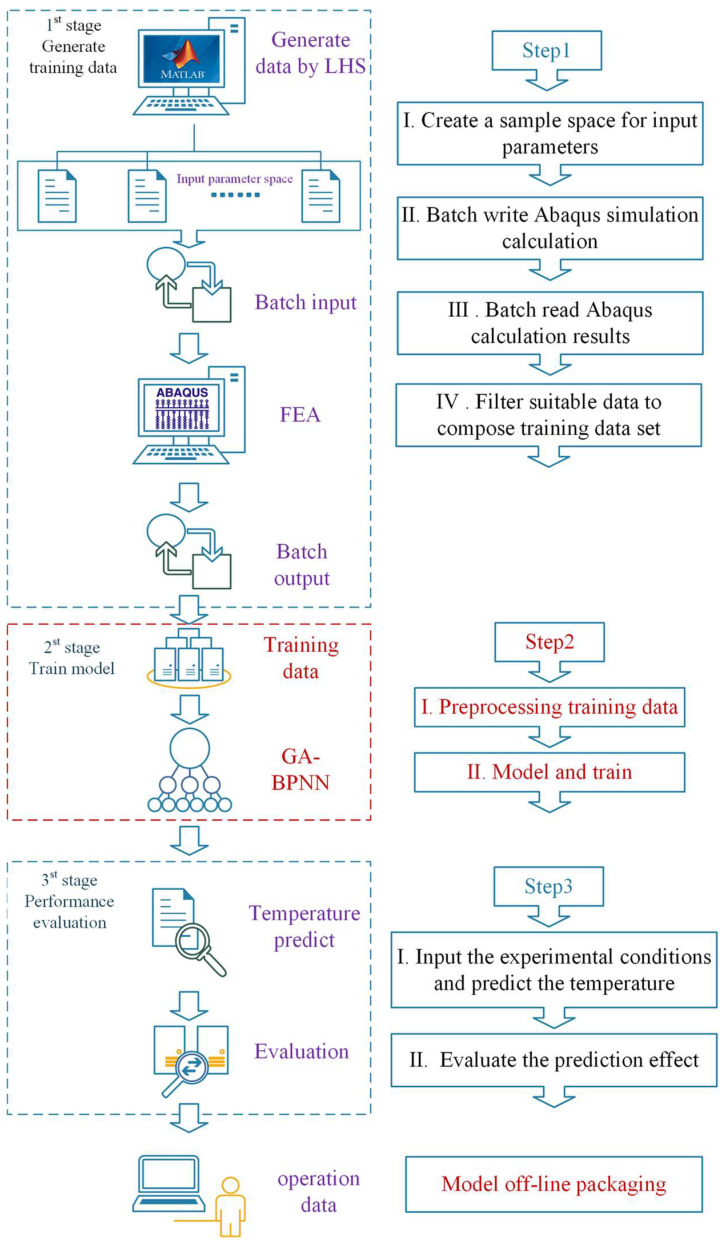
Construction process of temperature prediction model based on GA-BP neural network.

**Figure 7 micromachines-15-00430-f007:**
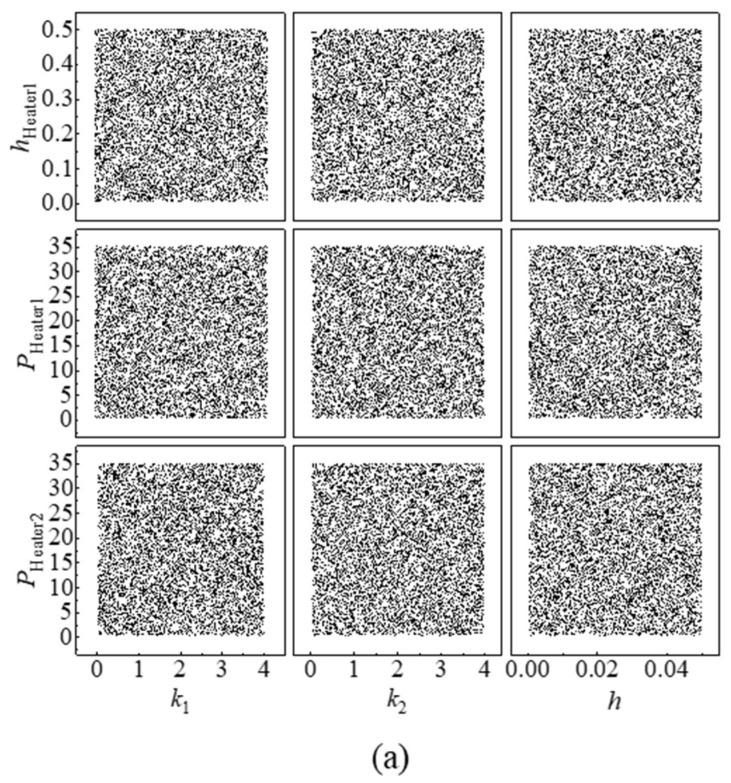
Distribution of partial parameters. (**a**) Distribution of six input parameters in the initial training data; (**b**) distribution of six input parameters in the modified training data, the red box represents the data that is filtered out.

**Figure 8 micromachines-15-00430-f008:**
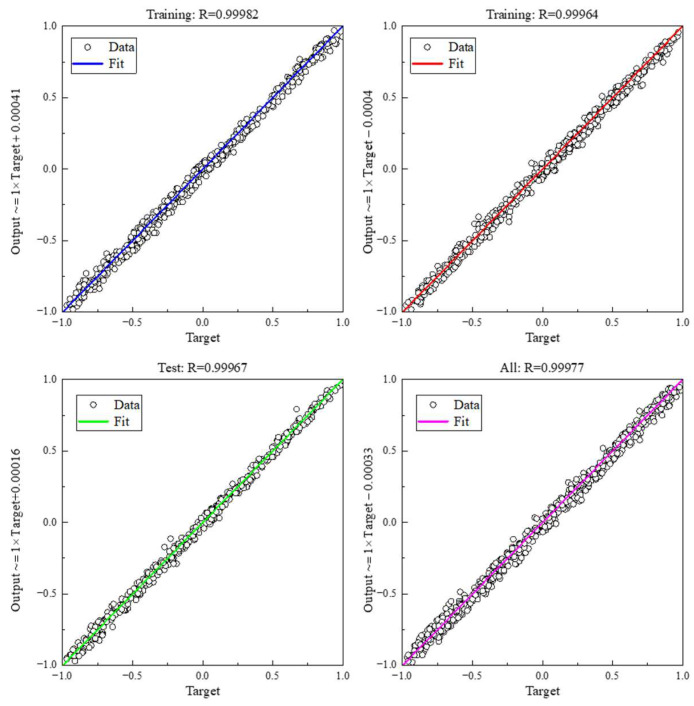
Regression of the pretraining model in the source domain.

**Figure 9 micromachines-15-00430-f009:**
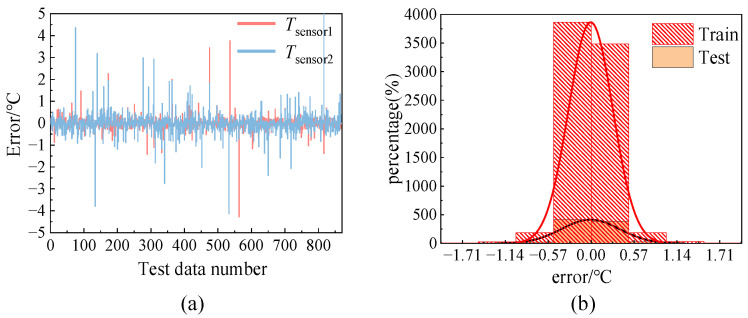
Output parameter errors distribution statistics. (**a**) the error of the output parameters from the test set; (**b**) *T*_sensor1_ error distribution of training set and test set.

**Figure 10 micromachines-15-00430-f010:**
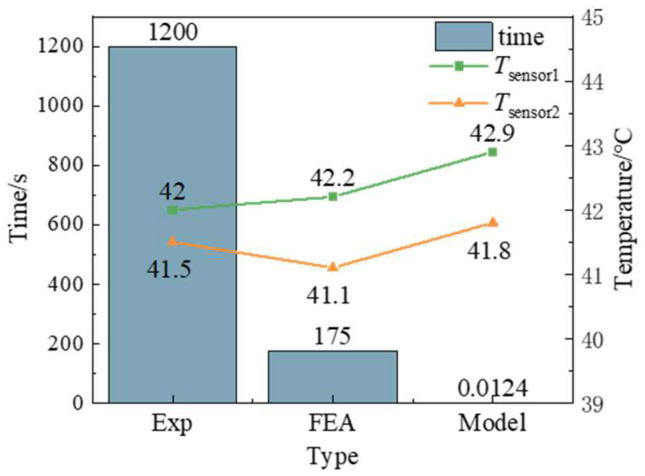
Comparison of the results of three type methods.

**Figure 11 micromachines-15-00430-f011:**
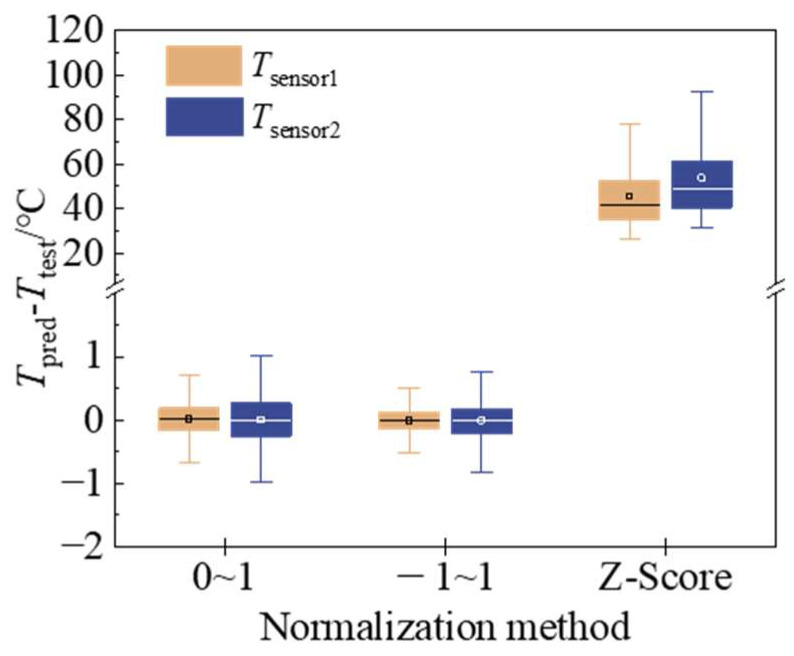
The influence of normalization method on model prediction error.

**Figure 12 micromachines-15-00430-f012:**
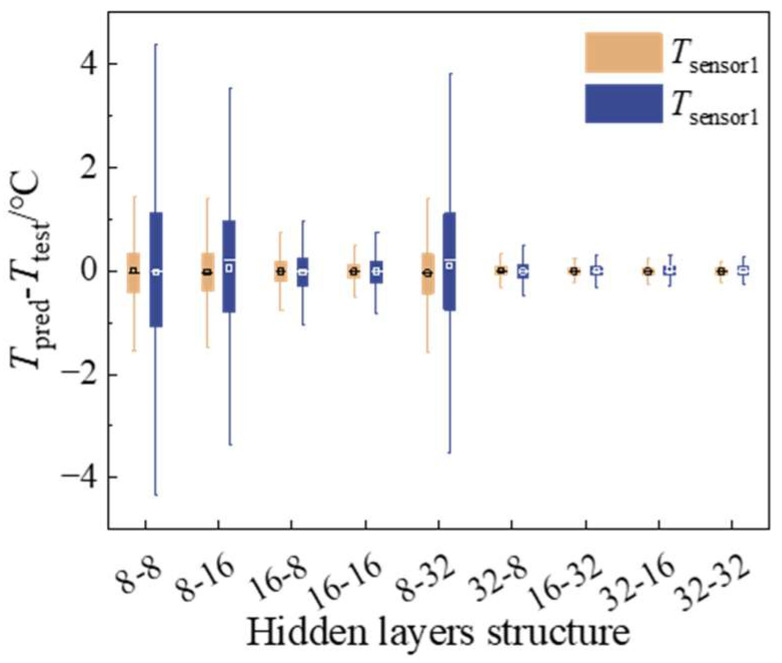
The influence of hidden layers structure on model prediction error.

## Data Availability

The data presented in this study are available on request from the corresponding author.

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
