# Peer review of "A Temperature Prediction Model for Flexible Electronic Devices Based on GA-BP Neural Network and Experimental Verification"

_micromachines, 2024, doi:10.3390/mi15040430_

Round 1
Reviewer 1 Report
Comments and Suggestions for Authors
In this paper, GA-BP neural network is used to predict the temperature of flexible electronic devices, and the predicted results are in good agreement with the experimental results, this study presents interesting features but minor revisions are required to make it worthy of publication in Micromachines.
1. The author sets the population size of the GA according to experience. Does this setting achieve the optimal performance of the GA? Were other population sizes set in the authors' experiments?
2. As described by the author in 3.2.2, the smallest mean and variance of the model error are present in the hidden layer structure of 16-32, Why was the hidden layer structure set to 16-16 in GA-BP model?
3. In the line 415, What does the “unable layer” in the text stand for?
Comments on the Quality of English LanguageThe quality of English language is basically fine.
Reviewer 2 Report
Comments and Suggestions for Authors
J. Nan et al., reported a method to assume the thermal conditions of the flexible electronic devices by establishing a BP neural network (GA-BPNN) temperature prediction model based on genetic algorithm optimization. This work claims that some critical challenges existing in conventional platforms – the high cost of experimental data input and computing powers are overcome by utilizing FEA software for simulating the input temperature data with different working conditions. The prediction achievements that the simulated variance of the GA-BPNN model was only under 0.57 degree C were commendable, and more importantly, it was successfully verified by transferring this to the estimation of thermal conditions in a flexible electronic device system. As the importance of the thermal-related issue of electronics is increasing, this work seems to have high scientific potential in this research field. Therefore, this manuscript is recommended for publication in this journal after addressing some minor comments below.
1. The flexible electronic device's design is simplified, possibly because of the convenience of their studies. However, as the device is becoming multifunctional by integrating many components in a single device, some more device parameters should be considered for realistic simulations that can be close to real applications. A soft encapsulation layer and/or filler of the flexible device will be critical for this kind of thermal estimation, and this can be briefly commented on in the manuscript.
2. In the introduction, some recent efforts in machine learning-based data acquisition of triboelectric-/piezoelectric sensors are introduced. Likewise, understanding sensor signals and the associated machine learning-based analysis are equally important in gas sensor applications. It is suggested to add some recent literatures about this (e.g. Journal of Hazardous Materials, 466, 133649, 2024 / ACS Nano 17, 539, 2023)
3. To further improve the thermal safety of the flexible electronics, this kind of simulation approach has a great synergy with other thermal safety engineering. For a more comprehensive understanding for the readers, some other strategies for enhancement in thermal safety should be cited (e.g. Nature Communications 14, 1024, 2023)
